# Data as a Consumable Resource

## Abstract

Data as an economic resource is increasingly relevant in an age of data-hungry parameterized models. The key property that sets classical data apart from traditional economic goods is the possibility of copying it at essentially no cost and without a record. The ability to store and transmit data as quantum states vulnerable to destructive measurement and unable to be generically copied has the potential to change this landscape. In this work, we use communication complexity lower bounds to demonstrate that encoding classical data in quantum states can mimic the behavior of a traditional, consumable economic good. We achieve this by proving that the quantum communication complexity of certain problems scales polynomially with the number of computational tasks performed using the data. This suggests that quantum networks hold the potential to enable novel types of data markets and incentive structures for the creation and distribution of *classical* data.

## 1. Introduction

Economic theory is concerned chiefly with goods that are consumed during the process of production, known as *rival* goods. However, for almost a century it has been recognized that data and information also play a vital role in economic processes (Schumpeter, 1942). The economic impact of data is particularly noticeable as statistical models fitted to large datasets are being usefully applied to problems in various fields of science and engineering (Brown et al., 2020; Chen et al., 2021a; Merchant et al., 2023). The ability to cheaply replicate data has long been recognized as its chief distinguishing characteristics compared to other economic resources, and this *nonrivality* has dramatic consequences (Arrow, 1962; Romer, 1990). It essentially implies that the (albeit idealized) equilibrium known as perfect competition,

in which the price of every good on the market is set by its capacity for increasing output, cannot hold once data is included. In some sense, one cannot "get their money's worth" when data is traded, unless there is some external enforcement mechanism that sets prices. Such a mechanism requires trust between the parties involved, and may have complex structure. Examples of this can be seen in recent proposals for data markets (Agarwal et al., 2018; Jones & Tonetti, 2020).

In contrast to the classical picture, the destructive nature of quantum measurement suggests that classical data encoded in the amplitudes of a quantum state may behave more like a traditional consumable economic resource, with potentially significant economic consequences [1]. For this to be the case, one must first show that a problem of interest can be solved with data encoded in this way. In addition, one must argue that the resulting states cannot be replicated in a similar manner to classical data. There is an inherent tension between these two goals, since while no-cloning is trivial for general states (Nielsen & Chuang, 2010), this is no longer the case once states are structured. As a simple example, given a computational basis state, it can be measured in the computational basis without disturbing it and copied in this way, and thus acts analogously to classical data. A less trivial example is states that encode boolean functions may also be copied in some cases due to their underlying structure (Aaronson, 2011). It is therefore a priori not obvious whether any problems satisfy these competing demands.

In Section 2 we make precise our definition of consumable quantum data. We then prove that for a number of problems, namely (i) sampling from a distribution defined by the solution of a linear system (Montanaro & Shao, 2022) (Section 3.1), (ii) a computational problem over bipartite graphs known as hidden matching (Bar-Yossef et al., 2008) (Section 3.2), and (iii) a multi-party two-outcome observable estimation problem (Appendix H), quantum data in fact behaves like a consumable resource. We also prove that classical data cannot behave in this manner. The main ingredient in the proofs is a lower bound on the quantum communication required to solve multiple instances of these problems when a subset of the inputs are correlated. The essence of the construction is that by sending a quantum

---

[1]Anonymous Institution, Anonymous City, Anonymous Region, Anonymous Country. Correspondence to: Anonymous Author <anon.email@domain.com>.

Preliminary work. Under review by the International Conference on Machine Learning (ICML). Do not distribute.

---

[1]For a very brief introduction to quantum mechanics, see Appendix A.

message, Alice in some sense reveals only a tiny fraction of her data while allowing Bob to solve these problems, and the method by which he solves them destroys the data and prevents it from being reused.

Motivated by the setting of data markets, we consider problems where one party (Alice) wants to sell some data in her possession to another party (Bob), while Bob is interested in evaluating a set of $m$ functions of Alice's data. Our results indicate that for certain problems, if Alice sends a Bob enough (classical) bits to evaluate a single function of interest, he can evaluate other functions as well, implying that Alice's potential payoff from selling her data will be independent of the number of Bobs (as for example in (Nageeb Ali et al., 2020)). If, on the other hand, Alice is willing to send only quantum copies of her data, Bob is in some sense forced to purchase a copy of the data state for each function he wishes to evaluate, and Alice's payoff may now be proportional to the number of functions. This is generally impossible to achieve using classical resources alone and without external mechanisms regulating the data market. We illustrate a possible application of our results by formulating a model of a data market as a strategic game in Section 4, in which Alice's payoff scales with the number of times her data is used when she uses quantum communication, but not when classical communication is used. We survey related work in Appendix F.

## 2. Consumable data

We now define the notion of consumable data. Denote by $\mathcal{X}, \mathcal{Y}, \mathcal{O}$ the space of Alice's inputs, Bob's inputs (or those of a single Bob in case there is more than one), and a space of outputs. Below, we use $P = (\mathcal{R}, \mathcal{P}_P, q)$ to denote a family of relational problems $\mathcal{R} \subseteq \mathcal{X} \times \mathcal{Y} \times \mathcal{O}$ and a set of protocols $\mathcal{P}_P$. We informally use problem to refer to tuples of this kind. We use $\mathcal{R}^m \subseteq \mathcal{X} \times \mathcal{Y}^m \times \mathcal{O}^m$ to denote the $m$-Bob relational problem where Alice receives one input and the $m$ Bobs have distinct inputs (or Bob receives $m$ inputs in the two-party case). The *goal* is to solve the relation on at least 60% of the Bob instances with 75% probability. Similarly, we use $\mathcal{P}_P^m$ to denote the set of protocols where Alice sends one message and the Bobs are allowed to communicate classically if $q = 0$ and quantumly if $q = 1$. So $P^m = (\mathcal{R}^m, \mathcal{P}_P^m, q)$.

For a problem $P$, denote $c(P)$ to be the communication complexity of the minimum cost protocol in the set $\mathcal{P}_P$ which solves $\mathcal{R}$. Since we will be modeling scenarios where Alice is selling her data to the Bobs who will be using it for computation, the cost here will be in terms of communication between Alice and the Bobs only.

**Definition 2.1.** A problem $P$ is said to be a *consumable data problem* if $c(P^m)/c(P) = \tilde{\Omega}(m^k)$ for some $k > 0$, and a *nonconsumable data problem* if $c(P^m)/c(P) = \tilde{O}(1)$

We refer to the quantity appearing in the lower bound in Theorem 2.1 as the *consumability factor* of $P$. In both these definitions, the tilde hides $\text{polylog}(m)$ factors. The motivation of these definition comes from the economic theory of production and the role of classical data in this framework, as we outline in Appendix G. There is a subtlety in this definition, in the sense that the benefit of consumability arises when Alice chooses to use a particular communication protocol (typically a quantum one over a classical one) but the definition itself does not specify why she would have such a preference. There are a few cases where consumability or nonconsumability can be characterized in some generality, which we consider in Appendix C.

## 3. Examples of consumable quantum data

### 3.1. Linear regression sampling

We consider here a sampling variant of linear regression introduced by Montanaro et al. (Montanaro & Shao, 2022). The notation in this section is defined in Appendix B.

*Problem* 1 (Multiple Linear Regression Sampling ($\text{MLRS}_{N,m}$)). Alice is given a vector $x \in \mathbb{S}^{N-1}$. Bob is given $m$ matrices $B_k$. The goal is for Bob to produce one sample from each distribution $\mathcal{P}_k$ over $[N]$ defined by $p_i^{(k)} = \left| \left[ B_k^+ x \right]_i \right|^2 / \left\| B_k^+ x \right\|_2^2$.

Note that solving the above problem with some inaccuracy $\eta$ corresponds to sampling from some distribution with total variation error at most $\eta$ with respect to $\mathcal{P}_k$. In order to consider the communication complexity of these problems, we must first discretize the inputs so that they have finite size. We thus assume all real number are specified to $\log N$ bits of precision.

**Lemma 3.1.** *i) For TV error $\eta \leq 1/4$, the quantum one-way communication complexity is*
$$SQ_\eta^\rightarrow(\text{MLRS}_{N,m}) = \Omega(m \log(N/m)).$$

*ii) For constant TV error $\eta$, in the multi-party setting, the one way quantum communication complexity is $SQ_\eta^\rightarrow(\text{MLRS}_{N,m}) = O(m \log(N) \max_k (\left\| B_k^+ \right\|^2 / \left\| B_k^+ x \right\|_2^2)).$*

Proof: Appendix E.

While these upper and lower bounds match in terms of their $N$ dependence if $\left\| B_k^+ x \right\|_2$ is relatively large (and in particular does not decay with $N$), they do not match in terms of their $m$ dependence. One example is when the features of $x$ that different samples are sensitive to are in some sense uniformly distributed, as in the construction used to obtain the lower bound in Theorem 3.1. In this case, we have $\max_k \left\| B_k^+ \right\|^2 / \left\| B_k^+ x \right\|_2^2 = O(m)$. In this setting, based on the definitions of Section 2, we obtain that

$\text{MLRS}_{N,m}$ is a consumable data problem for quantum data, with consumability factor $m$.

### 3.2. Hidden Matching

We define the following generalization of the Hidden Matching problem (Bar-Yossef et al., 2008):

*Problem* 2 (Multiple Hidden Matchings ($\text{MHM}_{N,m}$)). Alice is given a string $x \in \{0,1\}^N$. Each of the $m$ Bobs is given $m$ perfect matchings $\{M_k\}$ over $[N]$. Their goal is to output $(i, j, x_i \oplus x_j)$ where $(i, j) \in M_k$ for all $k$. Only Alice is allowed to send messages to Bob.

There is a quantum algorithm that solves this problem with probability 1 using $m \log N$ qubits of communication (Bar-Yossef et al., 2008): Alice sends Bob a copy of the state $|\psi\rangle = N^{-1/2} \sum_{i=1}^{N} (-1)^{x_i} |i\rangle$ over $\log N$ qubits. Denoting the $k$-th pair in Bob's a matching that Bob holds by $(i_k, j_k)$, Bob measures the state using the $N$-outcome POVM defined by $E_{k,b} = \frac{1}{2} \left( |i_k\rangle + (-1)^b |j_k\rangle \right) \left( \langle i_k| + (-1)^b \langle j_k| \right)$ for $k \in [N/2], b \in \{0,1\}$. This process is repeated for every matching. It is clear that the state cannot be re-used after such a measurement to solve the problem for multiple matchings. Since each POVM has $N$ possible outcomes, approaches based on gentle measurement that are discussed in Appendix H should not be applicable to this problem without requiring $\text{poly}(N)$ copies of the state.

We also have the following lower bound on the quantum communication required to solve the problem:

**Lemma 3.2.** $Q^{\rightarrow}(\text{MHM}_{N,m}) = \Omega(\sqrt{m})$ *for* $m \leq N/2$.

Proof: Appendix E

The combination of Theorem 3.2 and the upper bound implied by the algorithm described earlier implies that $\text{MHM}_{N,m}$ is a consumable data problem when quantum data is used, with consumability factor $\Omega(\sqrt{m}/\log(N))$.

### 3.3. Economic onsequences of consumable data

The fact that quantum data is consumable for both of these problems has consequences in a setting where Alice is interested in maximizing her profit when selling her data to Bob, who is interested in using it for computation. In this scenario, our lower bounds imply that if Alice chooses to use quantum communication, Bob must receive a message of size that is polynomial in the number of instances of the problem which he wants to solve. This is essentially because he cannot reuse the quantum states for solving multiple instances. This is in stark contrast to the picture when classical communication is used, since known bounds imply immediately that classical data exhibits nonrival behaviour for these problems (Appendix D). Once Alice sends Bob a message sufficient for solving a single instance, he can reuse it to solve multiple instances. We make these notions

more precise in the context of a strategic game that models a data market in Section 4.

When considering estimation of two-outcome observables, we show in Appendix H that quantum data is non-consumable in the two-player case, but is consumable in a variant with multiple Bobs that can only communicate classically and have limited memory.

## 4. A posted price data auction with consumable data

We would like to identify more concretely the economic consequences of the consumable nature of quantum data, using the linear regression sampling problem Section 3.1 as an example. We consider a formulation naturally related to auction theory (Krishna, 2009; Roughgarden, 2016). Alice's action space $A_A = \mathbb{R}_+$ is the set of prices she charges for a single bit or qubit of her input. Once Alice fixes a price $p$, Bob is free to purchase as many bits/qubits as he wants. Bob's action space is thus $A_B = \mathbb{N}$, and we denote the number he purchases by $b$. This is known as a posted price auction with only a single bidder and multiple items (or a particularly simple combinatorial auction). Assume the number of matrices Bob holds $m$ takes values in $[\overline{m}]$ and Alice has no knowledge of it (say she holds a uniform prior). We also assume the matrices $B_i, i \in [m]$ are chosen in a worst-case fashion (in order for our communication lower bounds to be applicable).

For any values of $m, p, b$, the payoffs of the two players are

$$v_A(m, p, b) = pb, \quad v_B(m, p, b) = \#\text{S}(m, b) - pb, \quad (4.1)$$

where $\#\text{S}(m, b)$ represents the number of samples Bob can produce using a message of $b$ bits/qubits, given that he holds $m$ such $B_i$).

Consider first the quantum communication case. We know from our lower bound Theorem 3.1 that for sufficiently large $m$, there is an absolute constant $C$ such that, if Bob were to purchase $b$ qubits produced by Alice, then

$$\#\text{S}^Q(m, b) \leq \frac{Cb}{\log(N/\#\text{S}^Q(m,b))} \approx \frac{Cb}{\log(N)} \quad (4.2)$$

for some absolute constant $C$. We also assume $N \gg m$ which allows us to use the approximation $\log N - \log \text{S}^Q(m, b) \approx \log N$ since this slightly simplifies the analysis. Since additionally $\#\text{S}(m, b) \leq m$ by definition, we have the upper bound

$$v_B^Q(m, p, b) \leq \min \left\{ \frac{Cb}{\log N}, m \right\} - pb. \quad (4.3)$$

If we also assume that Bob saturates this upper bound, to

maximize his payoff he solves

$$\max_b \min\left\{\frac{Cb}{\log N}, m\right\} - pb$$

$$= \begin{cases} m(1 - \frac{p\log N}{C}) & 0 < p < \frac{C}{\log N} & (b^\star = \frac{m\log N}{C}) \\ 0 & p \geq \frac{C}{\log N} & (b^\star = 0) \end{cases}$$

$$(4.4)$$

with the value of $b$ achieving the maximum in the right column. Alice's payoff is maximized by thus choosing $p$ as close as possible to $C$ from below without exceeding it, and will be equal to $b^\star(m,p)p = mp\log(N)/C$. This holds for any value of $m$ even though Alice has no knowledge of $m$, and in particular $C$ is independent of $m$.

In the classical case, we have tight upper and lower bounds on the number of bits needed to solve the problem (Theorem D.1), which give

$$v_B^C(m,p,b) = \mathbf{1}\left[b \geq C(N\log N)\right] m - pb. \quad (4.5)$$

For some constant $C'$. Bob thus solves

$$\max_b \mathbf{1}\left[b \geq C'N\log N\right] m - pb$$

$$= \begin{cases} m - pC'N\log N & p < \frac{m}{C'N\log N} & \left(\begin{array}{c} b^\star = \\ C'N\log N \end{array}\right) \\ 0 & p \geq \frac{m}{C'N\log N} & (b^\star = 0) \end{cases}$$

$$(4.6)$$

Note that unlike the quantum case, Alice has no way of knowing how to choose $p$ appropriately ahead of time, since it depends on $m$. If she wants to guarantee a nonzero payoff she has to choose $p = 1/(C'N\log N)$ (i.e. assume $m = 1$) in which case her payoff is independent of $m$ [2].

## 5. Discussion

We demonstrated that there exist problems for which encoding classical data into quantum states leads to behavior that is akin to that of rival, or consumable, goods, which is generally not possible using classical data alone. The inherent privacy benefits of amplitude-encoded data might also facilitate computation with proprietary data, giving users fine-grained control over the dissemination of their private data without the need for additional encryption. Being a preliminary investigation into the possibility of using quantum networks in this manner, our results do not immediately apply to problems with clear economic value. If this were the case, it could enable novel types of data markets and incentive structures for the production of data. It is worth noting however that our results for the linear regression sampling problem apply immediately to a related problem in which Bob obtains a state that encodes the solution to

---

[2] The expected payoffs (over the choice of $m$) do not exhibit any interesting dependence on the type of communication. They end up proportional to $\overline{m}$ in both the classical and quantum case.

a linear system rather than a classical sample. Such states are known to be strictly more powerful resources than classical samples (Aharonov & Ta-Shma, 2003), and could be useful in learning tasks such as updating the value of a linear estimator with new data (which is typically achieved with the recursive least squares algorithm). Another limitation of our work is that some of the bounds we present are loose, but these can possibly be tightened using more powerful tools such as Fourier analysis (Ben-Aroya et al., 2007). We discuss additional related directions of future work in Appendix I.

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

## A. A very brief review of quantum mechanics

We denote by $\{a_i\}$ a set of elements indexed by $i$, with 1-based indexing unless otherwise specified, with the maximal value of $i$ explicitly specified when it is not clear from context. $[N]$ denotes the set $\{0, \ldots, N-1\}$. The complex conjugate of a number $c$ is denoted by $c^*$, and the conjugate transpose of a complex-valued matrix $A$ by $A^\dagger$.

We denote by $|\psi\rangle$ a vector of complex numbers $\{\psi_i\}$ representing the state of a quantum system when properly normalized, and by $\langle\psi|$ its dual (assuming it exists). The inner product between two such vectors of length $N$ is denoted by

$$\langle\psi|\varphi\rangle = \sum_{i=0}^{N-1} \psi_i^* \varphi_i. \tag{A.1}$$

Denoting by $|i\rangle$ for $i \in [N]$ a basis vector in an orthonormal basis with respect to the above inner product, we can also write

$$|\psi\rangle = \sum_{i=0}^{N-1} \psi_i |i\rangle. \tag{A.2}$$

Matrices will be denoted by capital letters, and when acting on quantum states will always be unitary. These can be specified in terms of their matrix elements using the Dirac notation defined above, as in

$$A = \sum_{ij} A_{ij} |i\rangle \langle j|. \tag{A.3}$$

Matrix-vector product are specified naturally in this notation by

Quantum mechanics is, in the simplest possible terms, a theory of probability based on conservation of the $L^2$ norm rather than the standard probability theory based on the $L^1$ norm (Aaronson, 2017; Nielsen & Chuang, 2010). The state of a pure quantum system is described fully by a complex vector of $N$ numbers known as amplitudes which we denote by $\{\psi_i\}$ where $i \in \{0, \ldots, N-1\}$, and is written using Dirac notation as $|\psi\rangle$. The state is normalized so that

$$\langle\psi|\psi\rangle = \sum_{i=0}^{N-1} \psi_i^* \psi_i = \sum_{i=0}^{N-1} |\psi_i|^2 = 1, \tag{A.4}$$

which is the $L^2$ equivalent of the standard normalization condition of classical probability theory. It is a curious fact that the choice of $L^2$ requires the use of complex rather than real amplitudes, and that no consistent theory can be written in this way for any other $L^p$ norm (Aaronson, 2017). The most general state of a quantum system is a probabilistic mixture of pure states, in the sense of the standard $L^1$-based rules of probability. We will not be concerned with these types of states, and so omit their description here, and subsequently whenever quantum states are discussed, the assumption is that they are pure.

Since any closed quantum system conserves probability, the $L^2$ norm of a quantum state is conserved during the evolution of a quantum state. Consequently, when representing and manipulating quantum states on a quantum computer, the fundamental operation is the application of a unitary matrix to a quantum state. Given a quantum system with some discrete degrees of freedom, the number of amplitudes corresponds to the number of possible states of the system, and is thus exponential in the number of degrees of freedom. The simplest such degree of freedom is a binary one, called a qubit, which is analogous to a bit. Thus a state of $\log N$ qubits is described by $N$ complex amplitudes.

A fundamental property of quantum mechanics is that the amplitudes of a quantum state are not directly measurable. Given a Hermitian operator

$$\mathcal{O} = \sum_{i=0}^{N-1} \lambda_i |v_i\rangle \langle v_i| \tag{A.5}$$

with real eigenvalues $\{\lambda_i\}$, a measurement of $\mathcal{O}$ with respect to a state $|\psi\rangle$ gives the result $\lambda_i$ with probability $|\langle v_i|\psi\rangle|^2$. The real-valued quantity

$$\langle\psi| \mathcal{O} |\psi\rangle = \sum_{i=0}^{N-1} \lambda_i |\langle\psi|v_i\rangle|^2 \tag{A.6}$$

is the expectation value of $\mathcal{O}$ with respect to $|\psi\rangle$, and its value can be estimated by measurements. After a measurement with outcome $\lambda_i$, the original state is destroyed, collapsing to the state $|v_i\rangle$. A consequence of the fundamentally destructive nature of quantum measurement is that simply encoding information in the amplitudes of a quantum state dues not necessarily render it useful for downstream computation. It also implies that operations using amplitude-encoded data may incur measurement error, unlike their classical counterparts that are typically limited only by machine precision. The design of quantum algorithms essentially amounts to a careful and intricate design of amplitude manipulations and measurements in order to extract useful information from the amplitudes of a quantum state. For a comprehensive treatment of these topics see (Nielsen & Chuang, 2010).

## B. Notation

We denote by $D^\rightarrow$ deterministic classical one-way communication complexity. $R_\varepsilon^\rightarrow$ denotes randomized one-way classical communication complexity with error probability at most $\varepsilon$, in which players are allowed to share an unlimited number of random bits that are independent of their inputs. We similarly define by $Q_\varepsilon^\rightarrow$ one-way quantum communication complexity with error probability at most $\varepsilon$. In all cases the one-way restriction implies that only Alice is allowed to send messages to Bob (if there are multiple Bobs, they can communicate among themselves and we do not consider this as part of the complexity of the problem). When the error is a nonzero constant (say $1/4$) we omit the subscript.

We also consider sampling problems, where the goal is for Bob to produce a sample from a target distribution given some inputs to Alice and Bob. For this type of problem, we define analogously $SR, SQ$ for the classical (randomized) and quantum communication complexity respectively (with the superscript $\rightarrow$ denoting one-way communication as before). We allow constant error in TV distance between the target distribution and the one sampled by the algorithm. We denote by $A^+$ the pseudoinverse of $A$.

## C. Additional results on consumable data

Note that any problem involving only deterministic classical communication must be nonconsumable – every Bob can just copy Alice's message into his own working space. We also show in Appendix H that if $P$ corresponds to a *decision* problem, then even with quantum communication it must be a nonconsumable data problem. This is because the Bobs can apply the Shadow Tomography protocol (Aaronson, 2018) (unless the Bobs are only allowed classical communication between themselves and limited quantum memory). Nevertheless, consumability can be proved for certain search problems (with many solutions) solved using randomised or quantum communication.

**Lemma C.1.** *For any relational problem $\mathcal{R}$ and resource $q$, if the protocol is deterministic one way classical communication, $P = (\mathcal{R}, D^\rightarrow, q)$, then $\frac{c(P^m)}{c(P)} = 1$ and the data is nonconsumable.*

*Proof of Theorem C.1.* For the $m$-Bob problem, Alice sends the *same* message as the protocol for the original problem. Since her message depended only on her input, the message can be copied $m$ times and the correctness gaurantee holds for every instance on Bob's end. $\square$

Similarly,

**Lemma C.2.** *For any relational problem $\mathcal{R}$ and resource $q$, if the protocol is randomized one-way classical communication, $P = (\mathcal{R}, R^\rightarrow, q)$, then $\frac{c(P^m)}{c(P)} = 1$ and the data is nonconsumable.*

*Proof of Theorem C.2.* Akin to Theorem C.1, for every instance the success probability is at least 75% and thus with high probability 60% of Bobs succeed simultaneously. $\square$

**Lemma C.3.** *For any relation with an output space of size $K$, $\mathcal{R} \subseteq \mathcal{X} \times \mathcal{Y} \times \mathcal{O}$, with $|\mathcal{O}| = K$, if the protocol is one way quantum communication $P = (\mathcal{R}, Q^\rightarrow, q = 1)$ then $\frac{c(P^m)}{c(P)} = \widetilde{O}(K \log^2 m)c(P)$.*

*Proof of Theorem C.3.* Akin to Theorem C.2, we want to give a protocol for for $P^m$ using a protocol for $P$ as a subroutine. We do this by relying on the work of Gong, Aaronson (Gong & Aaronson, 2022) who proved that the distribution of $K$-outcome POVMs on $\log N$ qubits can be learned to constant additive error in $\widetilde{O}(K \log^2 m \log N)$ copies. $\square$

All of these lemmas can be generalised to the setting where $\mathcal{P}_P$ is a strict subset of one of these sets.

# D. Nonconsumability of classical data

## D.1. Linear regression sampling

Known results immediately imply the following bounds:

**Lemma D.1** ((Montanaro & Shao, 2022))**.** *For constant total variation distance error $\eta$ in the sampled distribution,*

   *i) The randomized one way classical communication complexity, $SR_\eta^\rightarrow(\mathrm{MLRS}_{N,1}) = \Omega(N \log N)$.*

   *ii) For any $m$, $SR_\eta^\rightarrow(\mathrm{MLRS}_{N,m}) = O(N \log N)$.*

*Proof of Theorem D.1.*   i) Theorem 9 of (Montanaro & Shao, 2022), applied to square matrices. The proof is based on lower bounds for distributed Fourier sampling.

  ii) It follows from the ability of Alice to send her whole input to Bob to complete the task.

## D.2. Hidden Matching

A tight lower bound shows that classical communication indeed acts like a nonrival good for this problem. While it is known that $R^\rightarrow(\mathrm{HM}_N) = \Omega(\sqrt{N})$ (Bar-Yossef et al., 2008), we believe this is the first characterization of the deterministic complexity of the Hidden Matching problem. The results are consistent with Theorem C.1.

**Lemma D.2.** $D^\rightarrow(\mathrm{MHM}_{N,m}) = D^\rightarrow(\mathrm{HM}_N) = N/2 + 1$.

*Proof of Theorem D.2.* We begin by proving $D^\rightarrow(\mathrm{MHM}_{N,1}) \geq N/2 + 1$.

A deterministic protocol $\mathcal{P}$ for $\mathrm{MHM}_{N,1}$ is defined by a matrix with $2^N$ rows denoting the inputs to Alice and $(N-1)!!$ columns denoting the inputs to Bob ($(N-1)!!$ is the number of perfect matchings over $[N]$). The entry in the matrix corresponding to inputs $(x, M)$ is a tuple $(i, j, b)$ such that $(i, j) \in M$ and $b = x_i \oplus x_j$. Define by $\tau$ a message sent by Alice, and by $S_\tau$ the subset of the rows for which Alice sends $\tau$ to Bob. The choice of $(i, j, b)$ depends on $x$ only through the message $\tau$. Since the protocol is deterministic, for a given column, the entries in each column of $S_\tau$ must have the same value since they share the same $\tau, M$, so we may write (with slight abuse of notation)

$$\mathcal{P}(x, M) = \mathcal{P}(\tau, M) = (i, j, b), \quad (i, j) \in M. \tag{D.1}$$

Thus the rows of $S_\tau$ are all identical, and we can view each entry as a constraint that each vector $x$ for which Alice sends the message $\tau$ must obey. We will bound the maximal possible size of $S_\tau$ by bounding the number of $x$s that can satisfy all these constraints.

The constraints on the bits can be thought of as edges on a graph $G = (V, E)$ with nodes $V$ indexed by $[N]$. We begin with $E = \emptyset$ and choose a sequence of matchings $\mathcal{M} = \{M^\ell\}$. For every matching, $\mathcal{P}$ must produce a valid output that selects an edge from the matching and constrains the corresponding entries of $x$. While we have no control over which edge is chosen, we will choose $\mathcal{M}$ in such a way that at each step of the algorithm, the size of the connected components in $G$ increases for any edge output by $\mathcal{P}$.

Denote by $\{C_i^\ell\}$ the connected components of $G$ at step $\ell$, and $C^\ell = \cup_i C_i^\ell$. Initially we thus have $\left|C^0\right| = 0$.

   i) $\left|C^\ell\right| \leq N/2$

      We start with an arbitrary matching $M^1$. For any $x$ for which Alice communicates $\tau$, the entries in $S_\tau$ in the column corresponding to $M^1$ is $S_\tau$ must contain an edge $(i, j) \in M^1$, hence after adding $(i, j)$ to $E$ and $M^1$ to $\mathcal{M}$ we have $\left|C^1\right| = 2$. Denoting by $D^\ell$ the disconnected nodes, we next define a matching $M^2$ that pairs each node in $C^1$ with some node in $D^1$. The remaining nodes of $D^1$ are paired among themselves. Note that $M^2$ cannot be equal to $M^1$, since $M^1$ contained an edge between two nodes that are both in $C^1$ while $M^2$ does not. We add $(i, j)$ to $E$ where $\mathcal{P}(\tau, M^2) = (i, j, b)$. If the edge connects $C^1$ and $D^1$, then $\left|C^2\right| = 3$. Otherwise, $\left|C^2\right| = 4$.

      We pick $M^3, \ldots$ in the same fashion, defining $M^{\ell+1}$ by pairing each node in $C^\ell$ with a node in $D^\ell$ (and pairing the remaining nodes arbitrarily). This can be done as long as $\left|C^\ell\right| \leq N/2$. At every stage, we are guaranteed that $M^{\ell+1} \notin \mathcal{M}$ by the same argument used for $M^2$, hence we are assured that it is a valid choice.

After at most $N/2 - 1$ such steps, we have either $|C^\ell| = N/2 + 1$ or $|C^\ell| = N/2 + 2$. From this point a different strategy is required, since there are not enough disconnected nodes in $D^\ell$ to pair with all the nodes in $C^\ell$. Subsequently, we order the nodes in $C^\ell$ by first ordering the connected components $\{C_i^\ell\}$ by size, with $C_0^\ell$ being the largest (or tied for the largest, breaking ties arbitrarily), and then arbitrarily ordering the nodes within each $C_i^\ell$.

ii) $|C^\ell| > N/2$ and $|C_0^\ell| \leq N/2$

Order the nodes in $C^\ell$ in the manner specified above. Denote by $R_-^\ell$ the first $N/2$ nodes in this ordering, and by $R_+^\ell$ the remaining $|C^\ell| - N/2$ nodes. Define the matching $M^{\ell+1}$ by first pairing each node in $R_+^\ell$ with a node in $R_-^\ell$ in descending order (i.e. starting with the nodes in $C_0^\ell$). Note that two nodes in the same connected component cannot be paired in this way. This is because, if this occurred for some connected component $C_i$, this would imply that either $|C_i| > |C_0^\ell|$ (since $C_i$ must have a node in $R_-^\ell$, the boundary between $R_-^\ell$ and $R_+^\ell$ divides $C_i$, so every node in the matching so far is in $C_i$, and we started the pairing in $R_-^\ell$ with the nodes in $C_0^\ell$ and went through all of them and reached $C_i$) contradicting the imposed ordering, or else $C_i = C_0^\ell$, in which case since some nodes in $C_0^\ell$ are also in $R_+^\ell$, we have $|C_0^\ell| > N/2$ and we terminate the algorithm. Having thus paired all the nodes in $R_+^\ell$ (we can always do this since $|R_+^\ell| \leq N/2$), we complete $M^{\ell+1}$ by pairing the remaining nodes in $R_-^\ell$ with the unconnected nodes $D^\ell$ in an arbitrary way. Note that $M^{\ell+1}$ does not contain any edge between two nodes that are in the same connectivity component. Thus it is distinct from all of the matchings already in $\mathcal{M}$ (since by construction each one contained such an edge) and we can add it to $\mathcal{M}$. We add $(j, k)$ to $E$ where $\mathcal{P}(\tau, M^{\ell+1}) = (j, k, b)$.

For the same reason specified above, the edge from $M^{\ell+1}$ that is selected by $\mathcal{P}$ will either connect two previously unconnected components in $C^\ell$ hence $C_k^{\ell+1} = C_i^\ell \cup C_j^\ell$ for some $i, j, k$, or else connect some $C_i^\ell$ with a previously unconnected edge (meaning $|C^{\ell+1}| = |C^\ell| + 1$).

We run the above algorithm until some step $\tilde{\ell}$ when either (a) $|C^{\tilde{\ell}}| = N$ or (b) $C_0^{\tilde{\ell}} > N/2$.

The algorithm is guaranteed to terminate in $O(N)$ steps. If (a) occurs, then either (a1) there are strictly less than $N/2$ connectivity components or (a2) there are exactly $N/2$ connectivity components, since each one contains at least two nodes. In case (a1), there are strictly less than $N/2$ independent degrees of freedom in the choice of the bits of any $x$ for which Alice sends the message $\tau$, since each connectivity component $C_i^{\tilde{\ell}}$ implies $|C_i^{\tilde{\ell}}|$ constraints of the form $x_j \oplus x_k = b$ where $\mathcal{P}(\tau, M) = (j, k, b), (j, k) \in M$ connects two nodes in $C_i^{\tilde{\ell}}$. In case (a2), there are $N/2$ connectivity components of size 2. We then consider a final matching $M^{\tilde{\ell}+1}$ that first divides $\{C_i^{\tilde{\ell}}\}$ into groups of two $\{K_i\}$ and then pairs each node to a node in a different connectivity component within the same $K_i$. As before, this matching is valid since $M^{\tilde{\ell}+1} \notin \mathcal{M}$. After including the edge in $\mathcal{P}(\tau, M^{\tilde{\ell}+1})$ into $E$, $G$ will contain $N/2 - 1$ connected components. As before, there are strictly less than $N/2$ degrees of freedom in choosing $x$. In case (b), there is a single component of size strictly larger than $N/2$. Thus even if all the remaining nodes are disconnected, there are strictly less than $N/2$ degrees of freedom once again.

In conclusion, in all cases we obtain that the number of rows of $S_\tau$ is at most $2^{N/2-1}$. The number of possible messages Alice must send is therefore at least $2^N / 2^{N/2-1} = 2^{N/2+1}$ and thus the number of bits Alice must send in order to solve $\mathrm{MHM}_{N,1}$ is at least $N/2 + 1$. Since this bound is valid for the multi-Bob version of the problem as well, we have $D^\rightarrow(\mathrm{MHM}_{N,m}) \geq N/2 + 1$.

The upper bound is trivial: Alice sends the Bobs the first $N/2 + 1$ bits of her input. These are sufficient for the Bobs to compute the output for all $m$ matchings simultaneously. The result follows. $\qquad\square$

$\qquad\qquad\qquad\qquad\qquad\qquad\qquad\qquad\qquad\qquad\qquad\qquad\qquad\qquad\qquad\qquad\qquad\qquad\square$

# E. Additional Proofs

*Proof of Theorem 3.1.* i) Say Alice is given a binary vector $y$ of length $m \log(N/m)$ and there are $m$ Bobs. Each Bob uses the matrix

$$B_j = \sum_{i=(N/m)j}^{(N/m)(j+1)} |i\rangle \langle i|. \tag{E.1}$$

Alice then divides her bits into $m$ sets of size $\log(N/m)$ and treats the bits in each set as an integer $r_j \in [N/m]$. She creates a vector $x$ of length $N$ by concatenating a unary encoding of these numbers, meaning

$$[x_{[(N/m)j:(N/m)(j+1)]}]_i = \sqrt{\frac{1}{m}} \delta_{ir_j}, \tag{E.2}$$

where we used $x_{[l:m]}$ denotes the subset of the entries of a vector ranging from $[l, m)$.

Suppose Alice and the Bobs manage to solve $\mathrm{MLRS}_{N,m}$ with inaccuracy $\eta$. This means that Bob produces a sample from a distribution that is at most $\eta$ in TV from each of his target distributions $\mathcal{P}_j$. From the definition of $x$ and the $B^{(j)}$, $\mathcal{P}_j$ is be a delta function at $r_j$. This means that with probability at least $1 - 2\eta$, Bob recovers the $\log(N/m)$ bits of $r_j$ by performing a computational basis measurement. It follows that Alice's message to Bob is a random-access encoding of $m \log(N/m)$ bits. From known lower bounds on the number of qubits needed for random access coding (Nayak, 1999), if $2\eta < 1/2$, Alice must send at least $\Omega(m \log(N/m))$ qubits to the Bobs.

ii) This follows immediately from the bound of Theorem 4 of (Montanaro & Shao, 2022) with an additional factor of $m$ due to the number of samples, and using $\|x\|_2 = 1$. The bound uses an amplitude-encoding of $x$, followed by the application of $B_k^+$ using block-encoding. If two-way communication is allowed, the complexity can be improved to $O(m \log(N) \max_k \|B_k^+\| / \|B_k^+ x\|_2)$ since Alice and Bob can run amplitude amplification.

$\square$

*Proof of Theorem 3.2.* Let us consider the distributional complexity of $\mathrm{MHM}_{N,m}$ where Alice's input is a uniform random string $\mathbf{X} \sim U(\{0, 1\}^N)$. The Bobs have a deterministic input Y, where $M_1$ is just the matching $\{(i, i + 1)|i \text{ odd}, i < N\}$. The matching $M_k$ is just the $k^{th}$ cyclic permutation on nodes on the left. The Bobs output random variables $o_k = (i_k, j_k, x_{i_k} \oplus x_{j_k})$ as their respective solutions. For notational convenience, we define $\mathbf{O} = o_1 o_2 \ldots o_m$. Note that since $m \leq N/2$, each matching consists of $N/2$ edges that do not appear in any other matching. It follows that for any choice of $\mathbf{O}$, no edge (as defined by the first two entries of each $o_k$) will be repeated.

Let $\rho_{\mathbf{X}}$ be density matrix corresponding to the message of length $l$ sent by Alice, of dimension $2^l$. By Holevo's theorem, $I(\mathbf{X} : \mathbf{O}) \leq l$. We will show that if the Bobs solve $\mathrm{MHM}_{N,m}$ then $I(\mathbf{X} : \mathbf{O}) \geq \Omega(\sqrt{m})$. This gives us the required lower bound.

$I(\mathbf{X} : \mathbf{O}) = H(\mathbf{O}) - H(\mathbf{O}|\mathbf{X})$. Note that $H(\mathbf{O}|\mathbf{X}) = 0$ since every Bob's output is deterministic given the input $\mathbf{X}$. Thus, $I(\mathbf{X} : \mathbf{O}) = H(\mathbf{O})$. To make this tuple amenable to analysis, we remove dependencies in the output by considering a spanning forest of the graph induced by $V = \cup_k \{i_k, j_k\} = \cup_k \{i_k\} \cup \cup_k \{j_k\}$. We have that $|V| \geq \sqrt{m}$ since we had a graph with $m$ distinct edges by construction. Therefore, we get a lower bound of $\Omega(\sqrt{m})$ by Theorem E.1. $\square$

**Lemma E.1.** *If we have a tree $T$ on $n$ vertices labelled with variables $x_1 \ldots x_n$, then if $x$ is a uniform random string then the set of random variables $P_T = \{b_{uv}|(u, v) \in T\}$ where each $b_{uv} = x_u \oplus x_v$ with probability at least 2/3 has total entropy at least $\Omega(n)$.*

*Proof.* We prove this by induction on the height of the tree $T$, say $h$. If $h = 0$, then we have only 1 vertex and the set of parities is empty so the entropy is 0. In the inductive step, we assume that for all trees of height $h - 1$, the statement is true. Now, consider any tree $T$ on $n$ vertices of height $h$. Let $L$ be the set of leaves of this tree, and set $k = |V(T) \setminus L|$. We know that the subtree of $T$ upto height $h - 1$ has total entropy on the set $P_{T,h-1}$ at least $k - 1$. For any vertex $v \in L$, let $p(v)$ be its parent. Then since $x_v$ is a uniform random bit and $v$ does not appear in any other parities, $H(x_v \oplus x_{p(v)}|P_{T,h-1}) = 1$. Since $b_{uv} = x_u \oplus x_v$ with probability 2/3, by concavity of entropy we have that $H(b_{uv}|P_{T,h-1}) = \Omega(1)$. We now iterate this argument over all leaves, adding the parities at the leaves to the conditioning. We thus prove our claim. $\square$

## F. Related Work

### F.1. Destructive measurement as a resource

The idea of using uncloneability of quantum states as a feature has a long history, starting with the seminal work of Weisner (Wiesner, 1983) that introduced the notion of quantum money. However, the states used in construction of quantum money schemes typically do not encode or transmit useful information and can benefit from the computational power of

pseudo-randomness in quantum state (Ji et al., 2018). While no-cloning is easy to show for states with little or no structure, this notion becomes more subtle for structured states, and in particular ones that might be useful in performing computation. Aaronson considered the question of uncloneablity of states that encode classical boolean functions, a problem known as quantum software copy-protection (Aaronson, 2011; Aaronson et al., 2020). He showed that the presence of structure enables such states to be cloned unless computational assumptions are made, and even then cannot be ruled out for states that encode functions that can be efficiently learned. The setting we consider can be seen as a distributed generalization of this problem. In the simplest case, evaluating the function of interest requires not only a quantum state in the possession of one player (or the equivalent classical description), but also an observable in the possession of another player.

**F.2. Communication complexity (classical and quantum)**

Our framework for demonstrating consumable behavior of data relies heavily on ideas from communication complexity (Yao, 1979). This is the study of distributed computational problems using a cost model that considers the communication required between parties. An excellent classical treatment is provided by (Kushilevitz & Nisan, 2011) (see also (Roughgarden, 2015)). The power of replacing classical bits of communication with qubits has been the subject of extensive study (Chi-Chih Yao, 1993; Brassard, 2001; Buhrman et al., 2009). Of particular relevance to our analysis are problems that exhibit exponential quantum communication advantages such as Hidden Matching (Bar-Yossef et al., 2008), Vector-in-Subspace (Raz, 1999) and sampling problems related to solutions of linear systems (Montanaro & Shao, 2022). See also (Nielsen & Chuang, 2010) for a general treatment of quantum computing.

# G. Data as an economic resource

Production theory (Kurz & Salvadori, 1995) is one of the principal frameworks for the quantitative study of economic systems. A fundamental object of interest within this framework is the *production function* $F : \mathbb{R}_+^M \to \mathbb{R}_+$ that quantifies in some form the output of an economic agent, for example the goods produced by a firm. The inputs to $F$ denote the resources required to produce said goods, such as labor, capital and raw materials. For conventional goods of this form, which cannot be replicated at zero cost (and are referred to as *rival* goods), it is known that the production function is typically a degree 1 homogeneous function of its inputs (at least locally when restricted to some set $S$):

$$F(\lambda x) = \lambda F(x) \tag{G.1}$$

for any $\lambda \geq 0$. This captures the notion that e.g. doubling the number of factories and raw materials will double a firm's output. It follows directly from Euler's theorem for homogeneous functions that within the interior of $S$,

$$F(x) = x \cdot \frac{\partial F}{\partial x}. \tag{G.2}$$

Since the output of the production function is a measure of the firm's capacity to pay for the needed resources, we see that if the price of resource $i$, denoted $p_i$, is set according to

$$p_i = \frac{\partial F}{\partial x_i}, \tag{G.3}$$

for all $i \in [M]$, then the output of the firm suffices exactly to purchase all the resources required, and there is no surplus profit. This is known as competitive equilibrium, which maximizes social welfare in the sense that the price of each good is commensurate to its usefulness in increasing the total output (Arrow, 1951; Debreu, 1959).

While it has long been understood at a qualitative level that data is an inherently different resource than the ones considered above due to the ability to copy it for free (Schumpeter, 1942; Arrow, 1962), the quantitative form of this statement was realized decades later by the seminal work of Romer (Romer, 1990). If we include data $y$ as an input into the production function, we instead have

$$F(\lambda x, y) = \lambda F(x, y) \tag{G.4}$$

rather than the expected need to double each input proportionate to match production as in $F(\lambda x, \lambda y) = \lambda F(x, y)$. This is because the data used by one process can be copied and used by several with negligible additional cost. Euler's theorem once again gives

$$F(x, y) = x \cdot \frac{\partial F}{\partial x}. \tag{G.5}$$

However, since increasing the amount of data will generally increase the output (say by improving the quality of inference), we have $\frac{\partial F}{\partial y} > 0$. It follows that

$$F(x, y) < x \cdot \frac{\partial F}{\partial x} + y \frac{\partial F}{\partial y}. \tag{G.6}$$

Due to this inequality, it is impossible to set prices according to Equation (G.3). If this were done for all inputs including data, the total output would be insufficient to pay for all the required resources. As a result, markets involving data must be inherently inefficient in the sense that one must underpay for some resource, or must include some external mechanism to enforce adequate compensation for resources that can be freely replicated. Mechanisms such as patent law or subsidies that incentivize innovation are all examples of this. Other examples are afforded by the trusted third parties that are introduced in proposals for data markets and handle the data in lieu of the data buyers themselves (Agarwal et al., 2018). In the context of strategic games that model data selling, the ability to copy data is also manifest in the payoff for the data seller being independent of the number of buyers, unless a mechanism is put in place by which the data buyers all agree to pay in advance for their data (Nageeb Ali et al., 2020).

The distributed problems we consider can be interpreted within this formalism. For example, the solution of $\text{MLRS}_{N,m}$ (Section 3.1) can be seen as the output of a production function, with the number of samples $m$ and Alice's message equivalent to $\lambda$ and $y$ respectively. The result of Theorem D.1 is then analogous to Equation (G.4). Up to constant factors, this is an example of the well-known nonrival nature of classical data. Alice must send a significant portion of her input to Bob for him to produce even a single sample, and once Alice sends her full input he can produce an unlimited number of samples in this way. If Alice were to sell Bob her data in the setting of a strategic game, her potential payoff will be essentially independent of the value that Bob can derive (since this is proportional to $m$).

On the other hand, Theorem 3.1 indicates that if Alice insists on using quantum communication, the data is analogous to a rival good as described by Equation (G.1). Bob can still produce $m$ samples, but this requires that Alice sends at least a number of qubits proportional to $m$. If Alice were to charge Bob for each qubit sent for example, she would obtain a payoff proportional to the Bob's output $m$ (as long as $m < N$). The lower bound indicates that this scaling holds regardless of the strategy Alice uses to encode her input into the message, and of the strategy Bob uses to process this message. Using classical resources alone this would be impossible to achieve.

The Multiple Hidden Matching problem (Section 3.2) and the associated bounds we prove can be interpreted similarly.

# H. Two-outcome observable estimation

In the previous sampling problem, we used the potential exponential quantum communication advantage in the linear sampling problem to make quantum representations of data a rival resource. It is then natural to ask if all problems exhibiting an exponential quantum advantage in communication can be made into a problem where data can be used as a rival resource. We will see that this is not the case quite generically, when Bob's task is a decision problem.

As a key example, consider the following problem built upon a classic problem in quantum communication complexity:
*Problem* 3 (Vector In Subspace ($\text{VS}_{N,\theta}$) (Kremer, 1995)). Alice is given a vector $x \in \mathbb{S}^{N-1}$. Bob is given two orthogonal subspaces of dimension $N/2$ specified by projection operators $M^{(1)}, M^{(2)}$. Under the promise that either $\left\| M^{(1)}x \right\|_2 \geq \sqrt{1 - \theta^2}$ or $\left\| M^{(2)}x \right\|_2 \geq \sqrt{1 - \theta^2}$ for $\theta < 1/\sqrt{2}$, their goal is to determine which is the case.

It is known that this problem exhibits an exponential advantage in quantum communication with respect to randomized classical communication complexity (Klartag & Regev, 2010). Consider the following generalization:
*Problem* 4 (Vector In Multiple Subspaces ($\text{VMS}_{N,\theta,m}$)). Alice is given a vector $x \in \mathbb{S}^{N-1}$. Bob is given $m$ pairs of orthogonal subspaces $M_j^{(1)}, M_j^{(2)}$. Given a similar promise to the vector in subspace problem for each pair of subspaces, the goal is to determine which subspaces $x$ has large overlap with.

The exponential advantage in quantum communication might suggest that for this problem as well, classical data will behave like a nonrival good while the quantum analog might behave like a rival good. Indeed the destructive nature of the measurements would naively seem to satisfy our intuitive requirements. This is because even for $m = 1$, Alice must send most of her input to Bob, and thus she may not be able to derive value that is proportional to $m$ for larger $m$. However, the problem can still be solved with relatively little quantum communication, since data states can be re-used in a manner that allows Bob to solve the problem for $m > 1$ with Alice communicating a number of qubits that is only logarithmic in $m$. This can be achieved via shadow tomography:

**Theorem H.1** (Shadow Tomography (Aaronson, 2018) solved with Threshold Search (Badescu & O'Donnell, 2020)). *For an unknown state $|\psi\rangle$ of $\log N$ qubits, given $m$ known two-outcome measurements $E_i$, there is an explicit algorithm that takes $|\psi\rangle^{\otimes k}$ as input, where $k = \tilde{O}(\log^2 m \log N \log(1/\delta)/\varepsilon^4)$, and produces estimates of $\langle\psi| E_i |\psi\rangle$ for all $i$ up to additive error $\varepsilon$ with probability greater than $1 - \delta$. $\tilde{O}$ hides subdominant polylog factors.*

$\text{VMS}_{N,\theta,m}$ is a problem of estimating $m$ expectation values up to some constant error (due to the constraint on $\theta$) on a target state. If polynomial error is required, it is known that $\Omega(N)$ qubits of communication may be required, and hence quantum communication is essentially equivalent to classical communication (from e.g. lower bounds on estimating inner products (Cleve et al., 1997)). Allowing constant error, Theorem H.1 implies that $\text{polylog}(m)$ qubits of communication suffice to solve the problem. This directly implies that, at least if Alice sends multiple copies of her state, a lower bound analogous to Theorem 3.1 is impossible, and Alice cannot hope to derive a payoff linear in $m$. This shows that an exponential communication advantage is not a sufficient condition for quantum data to behave like a rival good.

Given that multiple entangled copies of a quantum state are known to be a more powerful resource than single copies (Huang et al., 2022), it would also be interesting to consider a setting where Alice sends only single copies of her data states. One way to do this is by introducing assumptions about the computation Bob is allowed to perform with his message. Ideally this would be avoided, and one possible way to avoid this is using results on certified deletion (Broadbent & Islam, 2019). While requiring additional encryption, it may be possible in this way for Alice to only send a copy of her state after receiving a certificate that Bob has deleted the previous state, ensuring that multi-copy measurements cannot be performed.

A key difference between the vector in subspace problem and the other problems we consider is that the former is a decision problem (a two-outcome measurement), while the latter are sampling problems or relations. This may suggest that these type of problems are more amenable to the use of communication advantages to instill value in data as a resource.

### H.1. Multiple Bobs: An arms race?

Here we consider the task of two-outcome observable estimation more generally. If Alice has a vector which she can encode in a quantum state $|x\rangle$ and each of $m$ Bobs has an observable $O_i$, Alice is only willing to send the Bobs copies of $|x\rangle$ (when using quantum communication), and the Bobs cannot (i) store multiple copies of $|x\rangle$ or (ii) communicate quantum states between them, this is equivalent to the setting of learning without quantum memory that is studied in (Chen et al., 2021b). More precisely, this is a setting where each Bob can perform a POVM on a single copy of $|x\rangle$ only, and exchange classical messages which correspond to the classical memory used in this setting. In contrast, the setting of learning with quantum memory (as per (Chen et al., 2021b)) is one where the Bobs are allowed quantum communication (but still can measure only a single copy of $|x\rangle$ each), with the content of the quantum communication channel corresponding to the quantum memory. In both cases, Alice's messages correspond to samples of an unknown quantum state as is standard in learning problems. While the results of (Chen et al., 2021b) apply to samples of a mixed state described by a density matrix $\rho$, they also apply to a purification of $\rho$ in a larger space. This will not affect the scaling with $m$ which is the main object of interest for our purposes.

Define by $\mathcal{O}$ an ensemble of two-outcome POVMs given by $O_i = U_i Z_n U_i^\dagger$ for $0 \le i < m/2$ and $O_i = -U_{i-m/2} Z_n U_{i-m/2}^\dagger$ for $m/2 \le i < m$, where the $U_i$ are drawn i.i.d. from the Haar measure and $Z_n$ acts only on the last qubit.

When only classical communication is used between Alice and the Bobs, an optimal lower bound of $\Omega(\sqrt{N})$ for estimating the expectation value of a single two-outcome observable with constant probability is applicable (Gosset & Smolin, 2018). Lemma 1 of that paper also provides a matching upper bound in the $m$-observable case (up to logarithmic factors). Namely, estimating $m$ expectation values of unit norm observables to constant error can be done with probability $2/3$ by sending $\tilde{O}(\log(m)\sqrt{N})$ bits from Alice to Bob (where $\tilde{O}$ hides $\text{polylog}(N)$ factors). Alice requires no knowledge of the observables themselves. This protocol is based on sending $O(\log(m))$ random stabilizer sketches of Alice's input state $|x\rangle$. Each sketch involves Alice drawing a Clifford unitary $C$ from a uniform distribution over the Clifford group $\mathcal{C}_n$ ($n = \log N$), and computing $\langle 0^{\otimes(n-k)} z| C |x\rangle$ for all $z \in |0,1\rangle^k$ for $2^k = \tilde{O}(\sqrt{N})$. Alice generates $O(\log(m))$ i.i.d. sketches in this way and sends both the measurement results and a description of the Clifford unitaries to the Bobs. Each Clifford unitary is defined by specifying $O(n^2)$ one or two-qubit gates from a small set, and thus has an efficient classical description.

If Alice instead sends copies of her input encoded in the amplitudes of a quantum state $|x\rangle$ to the Bobs, but we allow classical communication only between the Bobs, and restrict the Bobs to performing single-copy measurements, the number of samples of $|x\rangle$ required is linear in $m$ (Chen et al., 2021b):

**Theorem H.2** (Corollary 5.7, (Chen et al., 2021b)). *With constant probability over $O_i$ drawn i.i.d. from $\mathcal{O}$, estimating the*

*expectation values of all $O_i$ w.r.t. $|x\rangle$ without quantum communication between Bobs with success probability at least $2/3$ requires $\Omega\left(\min\{m/\log(m), N\}/\varepsilon^2\right)$ copies of $|x\rangle$.*

Note that this is worst-case over $|x\rangle$ (if $|x\rangle$ was uniformly random Bobs could just guess 0). Note also that the $O_i$ are chosen so that classical shadows do not help (for the operators in question the Hilbert-Schmidt norm is $||O_i^2|| = N$, which is roughly equivalent to the shadow norm that sets the sample complexity of classical shadows (Huang et al., 2020)). A matching upper bound (up to $\log(m)$ factors, as long as $m < N$) is obtained by the straightforward approach in which Alice sends each Bob $O(1/\varepsilon_2)$ copies of her state.

When the Bobs are allowed to use quantum communication, they can jointly use shadow tomography (Aaronson, 2018; Badescu & O'Donnell, 2020) to estimate all the expectation values using a logarithmic number of copies of $\rho$.

These results are summarized in Table 1.

|  | Qubits/bits sent from Alice to the Bobs |
| --- | --- |
| Classical A → Bs, Classical Bs ↔ Bs | $\tilde{\Theta}(N^{1/2})$ (Gosset & Smolin, 2018) |
| Quantum A → Bs, Classical Bs ↔ Bs | $\tilde{\Theta}(m \log N)$ (Huang et al., 2020) |
| Quantum A → Bs, Quantum Bs ↔ Bs | $O((\log(m)\log(N))^2)$ (Badescu & O'Donnell, 2020) |

*Table 1.* A communication arms race in estimating expectation values of two-outcome observables to within constant error: Alice must send a significant portion of her input when using classical communication. If Alice sends quantum states and the Bobs use classical communication among themselves, Alice can obtain a large payoff (linear in $m$). If the Bobs also use quantum communication, she can no longer obtain such a payoff. We assume Alice only sends copies of her state, and assume constant error. $\tilde{\Theta}$ hides factors of $\log m$.

## I. Future work

Our results suggest that consumability arises most naturally in relational and sampling problems. Note that both of these classes of problems exhibit unique properties not shared by decision problems, which are the primary objects of study in complexity theory (Montanaro, 2019; Aaronson et al., 2023). For decision problems, we were able to demonstrate that the consumable data property only holds in the multi-Bob setting with classical communication between the Bobs. It is thus natural to ask whether consumability is unachievable for decision problems in the two-party setting. In this context, it is worth noting that our proof techniques are based on communication complexity reductions. These are cooperative problems, and it could be that leveraging Alice's ability to choose a data encoding that allows Bob to solve a problem yet somehow makes the task of copying the state more difficult, perhaps by additionally using cryptographic primitives, will change this picture.

The problems considered serve as a model for performing computation with user data, where both the data and the model used for computation are proprietary, valuable, and private. The setup we consider does not require end-users to possess a quantum computer in order to be valuable. Instead, the user must simply trust an entity possessing a networked quantum computer to distribute data states on their behalf (but not to perform more complex functions such as setting prices in a data market). This is similar to entrusting a bank to distribute funds on the behalf of an account holder.

The form of the quantum communication lower bound that indicates the rival behavior of quantum data is reminiscent of a direct sum theorem. Direct sum theorems demonstrate that the complexity of solving $m$ independent instances of certain problems scales linearly with $m$. They have been studied extensively in both the classical (Shaltiel, 2003; Braverman et al., 2013; Lee et al., 2008) and quantum (Sherstov, 2011; Jain & Kundu, 2020) setting. These results are not directly applicable since in our setting the inputs to Alice are not independent. Thus, this work motivates an *asymmetric* direct sum result for classes of communication relations.

### I.1. Task clonability and the possibility of computationally consumable data

In analogy to the potential clonability of quantum states with structure, there is a sense in which any non-consumable data may be cloned with respect to a particular task sample efficiently, even when cloning the overall state containing the information remains sample inefficient. This is exemplified by the shadow tomography task above in which the task is solved via the creation of a classical representation of a hypothesis $\rho_T$, such that $\tilde{r}(E_i\rho_T) \approx \tilde{r}(E_i\rho)$ for all $i$ for the ground truth state $\rho$. This classical representation $\rho_T$ need not be close in trace distance such that $||\rho - \rho_T||_{\text{tr}}$ is small, as

would be required for a high fidelity cloning of the true state. However it suffices for the task of shadow tomography, and admits an entirely classical representation that may be cloned through classical communication at will, making the data non-consumable, hence this task is clonable even when the underlying states might not be.

A lens we have not yet considered that is touched upon by the task of shadow tomography, is if data can become consumable when Bob is restricted in computational capacity. It has been noted that general shadow tomography procedures are expected to scale polynomially with the dimension of the Hilbert space of $\rho$ or the trial state $\rho_T$. If Bob is restricted to polylog computational time, then the creation of the clonable hypothesis state may become impossible. This is analogous to the effect in cryptographic no-cloning theorems on pseudorandom quantum states (Ji et al., 2018), where even when sample efficient cloning is possible, no computationally efficient scheme can be used to clone the states of interest. The addition of computational restrictions on Bob hence potentially widens the class of consumable data tasks, but requires moving beyond a purely commununcation model that permits unbounded computation. We leave this exciting area for future work.

### I.2. On purposeful data hiding - classical and quantum

So far we have discussed cases where quantum communication advantages along with the destructive nature of measurement naturally lead to situations where quantum data behaves as a rival good. In all these situations, however, the results are based on properties of the communication complexity of the problems, with the consequences for data economy was a byproduct. This is a convenient setting to understand how such an economy can arise naturally, but one may wonder if the introduction of cryptographic techniques can lead to additional situations where data may be treated as a rival good.

A setting that is known to bridge data privacy, destructive, and gentle quantum measurement is the study of differential privacy (Dwork & Roth, 2014; Aaronson & Rothblum, 2019). In differential privacy, a query is promised not to reveal too much about the specifics of the data, making it challenging to recreate and reuse the specifics of the data set by an individual user. Alternatively, one could consider the potential economic utility of primitives such as fully homomorphic encryption (Gentry, 2009) or blind quantum computation (Broadbent et al., 2008). Another interesting approach is to consider the dual view where rather than the data, the model is transmitted as a quantum state. This is of interest in practice, since both user data and the models used to compute with it are typically proprietary.

Moreover, in the above schemes we have focused on cases where the data is provided as pure states and when the only advantage examined otherwise is communication complexity. When data is transmitted as a pure state, it is known that classical shadows have the potential to remove strict communication advantages when the circuits are too simple, but reconstruction can still be challenging computationally based on arguments from quantum pseudorandom states (Zhao et al., 2023). An interesting direction could be to enhance some of the features of the example problems here by providing the data as a mixed state, such that it is more difficult for an adversary to learn under certain assumptions such as lack of a substantial quantum memory. We leave these directions for future work.

