# OpenReview forum: "Data as a Consumable Resource"
_ICML.cc/2024/Workshop/Agentic_Markets — Agentic Markets @ ICML'24 Poster_

### Official Review · Reviewer_Eg6b · 2024-06-10
**Review of Submission 10**

**Rating:** 6
**Confidence:** 3

**Review:**

**Quality**

This paper shows a connection between quantum mechanics and data markets, by deriving a type of consumable quantum data which arises in a distributed setting. This formulation allows for the authors to utilize communication complexity lower bounds to demonstrate that these quantum states function similarly to classical, consumable economic goods. The paper seems to be an initial foray into potentially using quantum data as an economic good within a distributed market, lending itself to applications utilizing ideas from classical economic theory. The framework and complexity theoretic arguments seem sound, and the subtleties which typically arise in quantum theory are well-explained, albeit in the appendices.

**Clarity**

The paper is clear in its writing and the quantum model and corresponding communication model are both clearly defined. However, it is less clear how widely applicable the results given are in a concrete sense. The paper first explores two examples of consumable quantum data (linear regression sampling and matching), but it is less clear how widely applicable the model is. Indeed, in the appendices it is shown that in a two-player strategic game, quantum data can be non-consumable. What are the failure modes/potential negative externalities of the model? And are there modifications to the decentralized quantum model that could be proposed?

**Originality**

While works that consider quantum states as a resource are relatively well-studied in the literature, this work tries to connect the non-cloneable nature of quantum states with the incentive structure of an economic resource. This is certainly an interesting and to my knowledge original interpretation which has many implications for future study.

**Significance**

In its existing state, the paper sets up the model and provides some indicative examples to argue for the quantum model they use as an economic resource. Conceptually, I believe the paper to be significant. However, more work needs to be done to properly understand the market externalities, as well as to find economically viable or useful applications of this framework. As it stands, I believe the paper to be worth accepting to ICML if more discussion about the limitations of the proposed model are added to the main text.

---

### Official Review · Reviewer_V8z2 · 2024-06-14
**Well-presented work on economic aspects of selling quantum data**

**Rating:** 8
**Confidence:** 3

**Review:**

**Summary**

This paper studies economic aspects of “quantum data” (encoding classical data in quantum states). The authors focus on the notion of data as a consumable resource (a resource such that consuming more costs more). Specifically, for several example problems, the authors prove (using communication complexity lower bounds) that quantum data has this consumable resource property (defined in terms of a certain “consumability factor”), while classical data does not.

The authors apply these results to a simple data selling scenario (in a simple fixed-price auction setting) – the data seller’s payoff scales linearly with the amount of data she sells in the quantum case, but remains constant in the classical case.

**Pros:**
- the paper reads nicely and the problem setting is well-motivated.

**Cons/Questions:**
- perhaps the authors could include more discussion in the main body of the paper regarding whether we should expect the consumability property to hold for other problems?

**Small comments:**
- Change running title for paper starting (currently reads as the default “Submission and formatting instructions for ICML”)
- L63: “Alice sends a Bob enough” → “Alice sends Bob enough”
- L56: “lower bound in Theorem 2.1” → “lower bound in Definition 2.1”